# Spatiotemporal dynamics and heterogeneity of renal lymphatics in mammalian development and cystic kidney disease

Daniyal J Jafree[1,2], Dale Moulding[1], Maria Kolatsi-Joannou[1], Nuria Perretta Tejedor[1], Karen L Price[1], Natalie J Milmoe[1], Claire L Walsh[3], Rosa Maria Correra[4], Paul JD Winyard[1], Peter C Harris[5], Christiana Ruhrberg[4], Simon Walker-Samuel[3], Paul R Riley[6], Adrian S Woolf[7,8], Peter J Scambler[1], David A Long[1]*

[1]Developmental Biology and Cancer Programme, UCL Great Ormond Street Institute of Child Health, University College London, London, United Kingdom; [2]MB/PhD Programme, Faculty of Medical Sciences, University College London, London, United Kingdom; [3]Centre for Advanced Biomedical Imaging, Division of Medicine, University College London, London, United Kingdom; [4]UCL Institute of Ophthalmology, University College London, London, United Kingdom; [5]Division of Nephrology and Hypertension, Mayo Clinic, Rochester, United States; [6]Department of Physiology, Anatomy and Genetics, University of Oxford, Oxford, United Kingdom; [7]School of Biological Sciences, Faculty of Biology Medicine and Health, University of Manchester, Manchester, United Kingdom; [8]Royal Manchester Children's Hospital, Manchester University NHS Foundation Trust, Manchester Academic Health Science Centre, Manchester, United Kingdom

*For correspondence:
d.long@ucl.ac.uk

Competing interests: The authors declare that no competing interests exist.

**Abstract** Heterogeneity of lymphatic vessels during embryogenesis is critical for organ-specific lymphatic function. Little is known about lymphatics in the developing kidney, despite their established roles in pathology of the mature organ. We performed three-dimensional imaging to characterize lymphatic vessel formation in the mammalian embryonic kidney at single-cell resolution. In mouse, we visually and quantitatively assessed the development of kidney lymphatic vessels, remodeling from a ring-like anastomosis under the nascent renal pelvis; a site of VEGF-C expression, to form a patent vascular plexus. We identified a heterogenous population of lymphatic endothelial cell clusters in mouse and human embryonic kidneys. Exogenous VEGF-C expanded the lymphatic population in explanted mouse embryonic kidneys. Finally, we characterized complex kidney lymphatic abnormalities in a genetic mouse model of polycystic kidney disease. Our study provides novel insights into the development of kidney lymphatic vasculature; a system which likely has fundamental roles in renal development, physiology and disease.

## Introduction

The lymphatic vasculature is a blind-ended network of vessels that is essential for tissue fluid balance, absorbance of dietary lipids and surveillance of the immune system in vertebrate homeostasis and has been implicated in cardiovascular disease (*Vieira et al., 2018*), neurodegenerative disorders (*Da Mesquita et al., 2018*), autoimmunity (*Louveau et al., 2018*) and cancer metastasis (*Le et al., 2016*). While the formation of systemic lymphatic vasculature, emerging from venous endothelium at

**eLife digest** In most organs in the body, fluid tends to build up in the spaces between cells, especially if the organs become inflamed. Each organ has a 'waste disposal system'; a set of specialized tubes called lymphatic vessels, to clear away this excess fluid and keep a check on inflammation. Defects in these tubes have been linked to a wide range of diseases including heart attacks, obesity, dementia and cancer.

The kidneys are responsible for filtering blood and balancing many of the body's chemical processes. Polycystic kidney disease (PKD) is the most common genetic kidney disorder and it results in cysts filled with fluid building up in the kidney. The growth of cysts in PKD may be due to a problem with the lymphatic vessels. However, compared to other organs, how lymphatic vessels first form within the kidney and what they do is not well understood.

Now, Jafree et al. have used three-dimensional imaging to study how lymphatic vessels form in the kidneys of mice and humans. The experiments showed that lymphatic vessels first appear when mouse kidneys are about half developed, and start to grow rapidly when the kidneys are thought to begin filtering blood. Clusters of cells that may help lymphatic vessels to grow were also found hidden deep within the kidneys of mouse embryos. Treating the kidneys with a factor that stimulates the growth of lymphatic vessels increased the numbers of these clusters. Jafree et al. found similar clusters of cells in human kidneys, suggesting that lymphatic vessels in the kidneys of different mammals may develop in the same way.

Further experiments showed that the lymphatic vessels of kidneys in mice with PKD become distorted early on in the disease, when cysts are still small and before the mice develop symptoms. In the future, identifying drugs that target kidney lymphatic vessels may lead to more effective treatments for patients with PKD and other kidney diseases.

around embryonic day (E)10 in mouse, has received considerable attention, how organ-specific lymphatics arise has only recently emerged as a focus of study, including the characterization of lymphatic development in the mouse mesentery (*Stanczuk et al., 2015*), intestine (*Mahadevan et al., 2014*), dermis (*Pichol-Thievend et al., 2018*; *Martinez-Corral et al., 2015*), meninges (*Antila et al., 2017*) and heart (*Klotz et al., 2015*; *Stone and Stainier, 2019*; *Maruyama et al., 2019*; *Gancz et al., 2019*). In contrast, how lymphatic vessels develop within the mouse or human kidney is not well established, despite their functional importance in pathological processes within the mature organ, including renal fibrosis and transplant rejection (*Sakamoto et al., 2009*; *Kerjaschki et al., 2004*; *Pei et al., 2019*).

The development of the metanephros, the direct precursor of the mature kidney, commences at around E10.5 in mouse and five post-conceptional weeks (PCW) in humans. Kidney development is a complex process, orchestrated by interactions between epithelial, stromal, immune and blood vascular endothelial cells (*McMahon, 2016*). How lymphatic vessels fit within this process is unclear. Our current understanding of kidney lymphatic development is limited to two descriptive studies; a paucity of insight which reflects their apparent scarcity (*Petrova and Koh, 2018*) and generally deep location within the kidney. Both prior studies draw conclusions based on staining tissue sections of rodent embryonic kidney with single lymphatic markers (*Lee et al., 2011*; *Tanabe et al., 2012*). However, the recent description of hybrid renal vessels with both blood vascular and lymphatic properties (*Huang et al., 2016*; *Kenig-Kozlovsky et al., 2018*) and non-endothelial renal cell types expressing lymphatic markers (*Hochane et al., 2019*; *Kim et al., 2015*) necessitate the simultaneous use of multiple markers to reliably identify lymphatic endothelium in the kidney.

Here, we have characterized the spatiotemporal development of the kidney lymphatic vasculature using a combination of wholemount immunofluorescence, optical clearing and high-resolution confocal microscopy to obtain three-dimensional (3D) images of intact mouse and human fetal kidneys at single-cell resolution. Using this approach, we provide novel insights into lymphatic vessel formation in mammalian kidney development, including a first description of the forming lymphatics in human fetal kidneys. Further, we identified lymphatic anomalies in a mouse model of polycystic kidney disease (PKD), the most common genetic cause of kidney failure.

## Results and discussion

### Spatiotemporal and quantitative dynamics of developing kidney lymphatic vessels in mouse

To observe lymphatic vessels during kidney development, we isolated metanephroi from CD-1 out-bred wildtype mouse embryos at a range of developmental stages, and wholemount immunola-belled them for the early lymphatic endothelial cell markers (*Banerji et al., 1999*; *Wigle and Oliver, 1999*) prospero homeobox protein 1 (PROX1) and lymphatic vessel endothelial hyaluronan receptor 1 (LYVE-1). We then optically cleared and imaged entire labelled kidneys using confocal microscopy. 3D reconstructions and z-stacks of confocal images revealed a cellular network of PROX1[+]/LYVE-1[+] lymphatic endothelial cells in the E14.5 mouse kidney (*Figure 1A*). At this stage in development, the ureteric bud, has already undergone 8–9 generations of branching (*Short et al., 2014*) and the kidney contains a rich, perfused blood vasculature (*Loughna et al., 1997*; *Munro et al., 2017*; *Rymer et al., 2014*). No definitive PROX1[+]/LYVE-1[+] vessels were observed in the kidney prior to E14.5 (data not shown). By E16.5, the kidney lymphatics constituted a vascular plexus, that contained some lumenized vessel segments (*Figure 1—video 1*). By E18.5, lymphatic vessel branches were observed distally in the kidney. We confirmed the lymphatic identity of the plexus (*Figure 1—figure supplement 1*) by its prominent expression of vascular endothelial growth factor receptor 3 (VEGFR-3) and podoplanin (*Breiteneder-Geleff et al., 1997*; *Kaipainen et al., 1995*) at E15.5 and weak expression of the blood endothelial marker endomucin relative to surrounding blood vasculature (*Stanczuk et al., 2015*; *Hägerling et al., 2013*).

Using a computational approach for the segmentation and volume rendering of PROX1[+]/LYVE-1[+] vessels, we created 3D models of lymphatic development in the mouse embryonic kidney (*Figure 1B*, and *Figure 1—video 2*). These models visually depict the progressive remodeling of kidney lymphatic vessels from E14.5 through to E18.5. At all stages we observed a ring-like anastomosis in the renal hilum, from which lymphatic vessels originated. These vessels branched and increased in length over the course of renal development. The 3D models conveyed that large volumes of the embryonic kidney were unoccupied by lymphatic vessels, demonstrating that kidney lymphatics are sparse compared to blood vasculature at equivalent stages (*Munro et al., 2017*; *Daniel et al., 2018*; *Sequeira Lopez and Gomez, 2011*). We then used uroplakin (UPK3A), platelet endothelial cell adhesion molecule-1 (PECAM-1), aquaporin 2 (AQP2) and alpha smooth muscle actin (αSMA) to clarify the spatial relationship of lymphatic vessels to other structures within the developing kidney (*Figure 1—figure supplement 2*). We found that the ring-like lymphatic anastomosis resided under the nascent renal pelvis; identified by the urothelial marker UPK3A (*Figure 1—video 3*). PECAM-1[+] blood endothelium and lymphatic vasculature run alongside each other in the kidney, as shown in other organs (*Mahadevan et al., 2014*; *Klotz et al., 2015*). Whereas αSMA was expressed around the PECAM-1[+] renal arterial system at E18.5, as previously described (*Pitera et al., 2004*), we did not detect αSMA around PROX1[+] vessels. This suggests that pre-collectors or collector vessels have not yet formed by this time-point or that lymphatics within the kidney are capillaries that lack mural cell coverage (*Wang et al., 2017*). We labelled collecting duct epithelium with AQP2 to delineate the renal medulla and found no lymphatic vessels in this region of the kidney (*Figure 1—figure supplement 2*). At no point were lymphatics observed near or within the renal capsule.

Quantitative analysis of 3D imaging has provided novel insights into mouse kidney development, such as the stereotypical nature of ureteric bud branching and the cellular dynamics of the progenitor niches that eventually form the nephron, the functional unit of the kidney (*Short et al., 2014*; *Short et al., 2018*; *O'Brien et al., 2018*). To perform the first quantitative 3D analysis of kidney lymphatic vessel development, we segmented and analyzed PROX1[+]/LYVE-1[+] vessels from entire mouse embryonic kidneys using filament tracing software to measure the number of kidney lymphatic vessel branches including lengths, mean diameters and volumes for each vessel branch. From these data, we quantified vascular parameters: the range of vessel branch lengths, the maximum vessel branch diameter and the total volume of all branches constituting the lymphatic network between E14.5 and E18.5 (*Figure 1C*). Overall, between E14.4 and E18.5, the total number of lymphatic vessel branches increased over five-fold (p=0.0003); the range of vessel lengths increased approximately three-fold (p<0.0001); the maximum diameter approximately doubled (p=0.003); and the total network volume increased by a factor of 18 (p=0.0006). During this time, the volume of the kidney expands over 40-fold (*Short et al., 2014*) and mature renal cell types emerge in the kidney

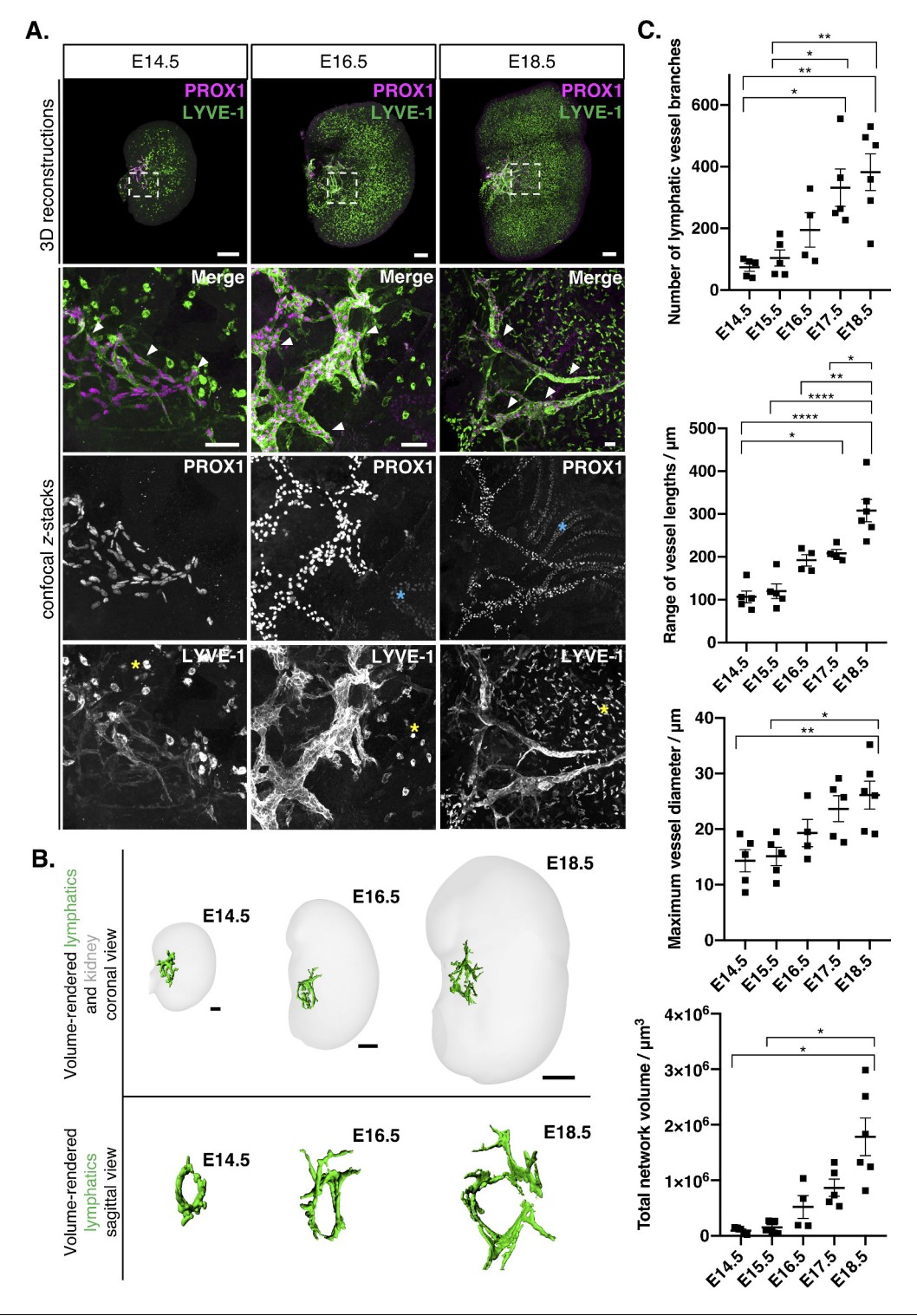

**Figure 1.** Visual and quantitative characterization of lymphatic vessels during mouse kidney development. (**A**) Appearance of PROX1[+]/LYVE-1[+] lymphatic endothelium (white arrowheads) in E14.5, E16.5 and E18.5 mouse kidneys. Representative 3D reconstructions and confocal z-stacks from a minimum of four kidneys per timepoint from three independent embryonic litters. Dashed boxes in the 3D reconstructions represent the location of the lymphatic plexus and indicate area shown at higher magnification in the bottom of each panel. Strong LYVE-1 staining was also observed in macrophages from E14.5 through to E18.5 (yellow asterisks). Weak PROX1 staining was observed in tubular epithelium from E16.5 onwards (blue asterisks). 3D reconstruction scalebars: 100 μm

*Figure 1 continued on next page*

*Figure 1 continued*

(E14.5, E16.5); 200 μm (E18.5). Confocal z-stack scalebars: 50 μm. (B) Volume renderings of the lymphatic plexus and embryonic kidneys at E14.5, E16.5 and E18.5; top panels show the lymphatic plexus and kidney from a coronal view; bottom panels show the sagittal view looking down upon the plexus towards the hilum. Scalebars: 50 μm (E14.5); 100 μm (E16.5); 500 μm (E18.5). The sagittal views are not to scale. (C) Quantitative analysis of kidney lymphatic vessel development at E14.5 (*n* = 5), E15.5 (*n* = 5), E16.5 (*n* = 4), E17.5 (*n* = 5) and E18.5 (*n* = 6). Each data point represents a whole kidney immunolabelled for PROX1 and LYVE-1 and error bars represent standard error of the mean. ANOVA showed significant increases in the number of lymphatic vessel branches per kidney (*F* = 8.5; p=0.0003), the range of vessel lengths (*F* = 20.6; p<0.0001), the maximum vessel diameter (*F* = 5.7; p=0.003) and total network volume (*W* = 10.6; p=0.0022). For the total volume of the lymphatic network, we used Welch's ANOVA, as equality of variance was not met for this parameter. Brackets between timepoints demonstrate significance when multiple comparisons were performed using Bonferroni tests (or Tamhane T2 multiple comparison testing for total network volume). *: p<0.0332; **: p<0.0021; ***: p<0.0002, ****: p<0.0001. Raw data and results of all multiple comparisons are presented in *Figure 1—source data 1*.

The online version of this article includes the following video, source data, and figure supplement(s) for figure 1:

**Source data 1.** This spreadsheet contains the numerical raw data and statistics for the quantitative analysis of kidney lymphatic vessel development shown in *Figure 1C* and *Figure 1—figure supplement 3*.
**Figure supplement 1.** Confirmation of identity of the kidney lymphatic plexus.
**Figure supplement 2.** Spatial relationships of lymphatic vessels to other anatomical structures in the developing mouse kidney.
**Figure supplement 3.** Filament tracing analysis for quantification of kidney lymphatic vessel development.
**Figure supplement 4.** Proximity of VEGF-C expression during murine kidney development.
**Figure 1—video 1.** Appearance of patent vessel lumen in the kidney lymphatic plexus at E16.5.
https://elifesciences.org/articles/48183#fig1video1
**Figure 1—video 2.** 3D imaging and volume rendering of the developing kidney lymphatic vasculature.
https://elifesciences.org/articles/48183#fig1video2
**Figure 1—video 3.** Anatomical relationship of the lymphatic plexus to the nascent renal pelvis.
https://elifesciences.org/articles/48183#fig1video3

including pericytes and mesangial cells (*Mugford et al., 2008*), podocytes and all the components of the differentiated nephron (*Combes et al., 2019*). We then compared kidneys at intermediate timepoints between E14.5 and E18.5. No lymphatic vascular parameters increased significantly between E14.5 and E15.5. In contrast, between E15.5 and E18.5, we observed a significant increase in the total number of vessel branches (p=0.0038), their range of lengths (p<0.0001), maximum diameter (p=0.0166) and total network volume (p=0.0429). There were further significant increases in the number of vessel branches between E15.5 and E17.5 (p=0.0322) and the range of vessel lengths between E16.5 (p=0.0039) or E17.5 (p=0.0154) when compared to E18.5.

By pooling values for lymphatic vessel branch parameters for all kidneys at each timepoint, we generated histograms conveying the increase in vessel number and branch length, diameter and volume at each timepoint. The change in shape of histograms between each timepoint were less pronounced between E14.5 and E15.5, suggesting rapid remodeling of the plexus after E15.5 (*Figure 1—figure supplement 3*). Together, our quantitative approach suggests a period of quiescence in the early stages of kidney lymphatic vessel development, followed by a progressive increase in lymphatic expansion after E15.5. This timing just precedes the initiation of excretory function by the kidney (*McMahon, 2016*; *Caubit et al., 2008*) and also coincides with the appearance of primitive erythrocytes in renal vasculature (*Munro et al., 2017*) and a wave of development of vascularized glomerular capillary loops (*Hu et al., 2016*).

The rapid expansion of the lymphatic plexus from E15.5 onwards could be driven by the accumulation of interstitial fluid, which has extravasated from the renal vasculature and stimulates lymphatic expansion (*Planas-Paz et al., 2012*). Additionally, either the recruitment of progenitor cells, as previously suggested for other organs (*Escobedo and Oliver, 2016*; *Kazenwadel and Harvey, 2016*), or cellular and paracrine factors may be responsible (*Vaahtomeri et al., 2017*). One critical paracrine factor is vascular endothelial growth factor C (VEGF-C), which binds to VEGFR-3 and promotes the sprouting and migration of lymphatic endothelial cells (*Pichol-Thievend et al., 2018*; *Hägerling et al., 2013*; *Karkkainen et al., 2004*). To explore the expression of VEGF-C in the developing kidney, we used mice carrying *LacZ* under the control of the endogenous *Vegfc* regulatory

region (*Karkkainen et al., 2004*) (*Figure 1—figure supplement 4*). Wholemount X-gal staining of E15.5 kidneys from heterozygous embryos (*Vegfc^LacZ/+^*) demonstrated β-galactosidase (β-gal) activity in a branching pattern, mimicking the appearance of the renal arterial tree. Sections of X-gal-stained E16.5 *Vegfc^LacZ/+^* kidneys showed β-gal activity to be restricted to interstitial cells beneath the pelvis and adjacent arterioles. We further stained these sections for LYVE-1 and observed lumenized LYVE-1$^+$ vessels in the hilum surrounded by β-gal-expressing interstitial cells. Together, these findings convey that the renal hilum; where kidney lymphatics first arise, is a VEGF-C-rich niche.

## Characterization of conserved lymphatic endothelial cell clusters in the developing mammalian kidney

During embryonic development and in the early postnatal period, lymphatics form by sprouting from veins and pre-existing lymphatics; a process termed lymphangiogenesis, and the assembly of lymphatic progenitors; a process termed lymphvasculogenesis (*Potente and Mäkinen, 2017*). A hallmark of lymphvasculogenesis is the presence of isolated clusters of lymphatic endothelial cells, as observed during the development of mesenteric, meningeal, dermal and cardiac lymphatic vasculature (*Stanczuk et al., 2015*; *Pichol-Thievend et al., 2018*; *Martinez-Corral et al., 2015*; *Antila et al., 2017*; *Stone and Stainier, 2019*; *Gancz et al., 2019*). By inspecting confocal image stacks of E16.5 mouse embryonic kidneys, we found PROX1$^+$/LYVE-1$^+$ cellular clusters, which were structurally distinct from the lymphatic vessel plexus (*Figure 2A* and *Figure 2—video 1*). We confirmed the lymphatic identity of these clusters by their expression of VEGFR-3 and podoplanin at E15.5 and E16.5 (*Figure 2B*). LYVE-1, VEGFR-3 and podoplanin all highlighted filopodia-like processes extending from lymphatic clusters in the kidney, likely analogous to the migratory tips that extend from nascent lymphatic endothelium (*Xu et al., 2010*). We performed further immunolabelling to characterize the molecular profile of the clusters (*Figure 2B*, *Figure 2—figure supplement 1*). The PROX1$^+$/LYVE-1$^+$ clusters did not express the murine macrophage marker F4/80 (*Munro et al., 2019*). Relative to renal blood vasculature, PECAM-1 and endomucin were weakly expressed by lymphatic clusters, supporting their non-blood vascular endothelial identity (*Podgrabinska et al., 2002*).

As our imaging technique captured entire mouse embryonic kidneys at single-cell resolution, we were able to use a quantitative approach to assess the dynamics of lymphatic endothelial cell clusters during kidney development. For each embryonic kidney, we counted the frequency of PROX1$^+$/LYVE-1$^+$ clusters and quantified the total number of cells constituting all clusters within each kidney (*Figure 2C*). Between E14.5 and E16.5 there was a significant increase in both the frequency of clusters (p=0.0001) and number of total cluster cells (p<0.0001). Both parameters peaked at E16.5 but after E16.5, both frequency (p<0.0064) and number of total cluster cells (p<0.0064) declined significantly. This decline coincides with a reduction in the generation of new ureteric bud branches and a wave of new nephron formation (*Short et al., 2014*), and agrees with the kidney lymphatic vessel expansion we observed after E15.5. We also assessed the morphology of clusters within mouse embryonic kidneys using 3D segmentation and volume rendering (*Figure 2D*). At each timepoint, we arranged volume-rendered clusters from any single kidney into a hierarchy, starting with isolated PROX1$^+$/LYVE-1$^+$ cells through to large islands containing eight cells or more.

To investigate whether VEGF-C might also be involved in the formation of kidney lymphatic clusters, we isolated and cultured intact E14.5 metanephroi ex vivo (*Figure 2E*). 3D imaging demonstrated scattered PROX1$^+$/LYVE-1$^+$ clusters in control metanephroi after 48 hr in culture, and their abundance increased significantly upon supplementation of the culture media with recombinant VEGF-C (p=0.0184; *Figure 2—video 2*). As multiple stages of cluster development exist within any single kidney and the decline in cluster frequency coincides with rapid lymphatic vessel expansion, we propose that both lymphangiogenesis and lymphvasculogenesis contribute to the formation of the mature kidney lymphatic vasculature and that both are regulated by VEGF-C.

Lymphatic development has been described predominantly in model organisms such as mouse and zebrafish (*Semo et al., 2016*). Prior studies addressing normal human lymphatic development (*von Kaisenberg et al., 2010*; *Cho et al., 2012*; *Jin et al., 2010*; *Belle et al., 2017*) have mostly examined systemic lymphatics and have only used single markers such as podoplanin. Thus, it is still not clear whether events observed in animal models can be extrapolated to lymphatic development in human organs. Moreover, human kidney development has been described and thoroughly compared with that of mouse (*Lindström et al., 2018a*; *Lindström et al., 2018b*), though the existence

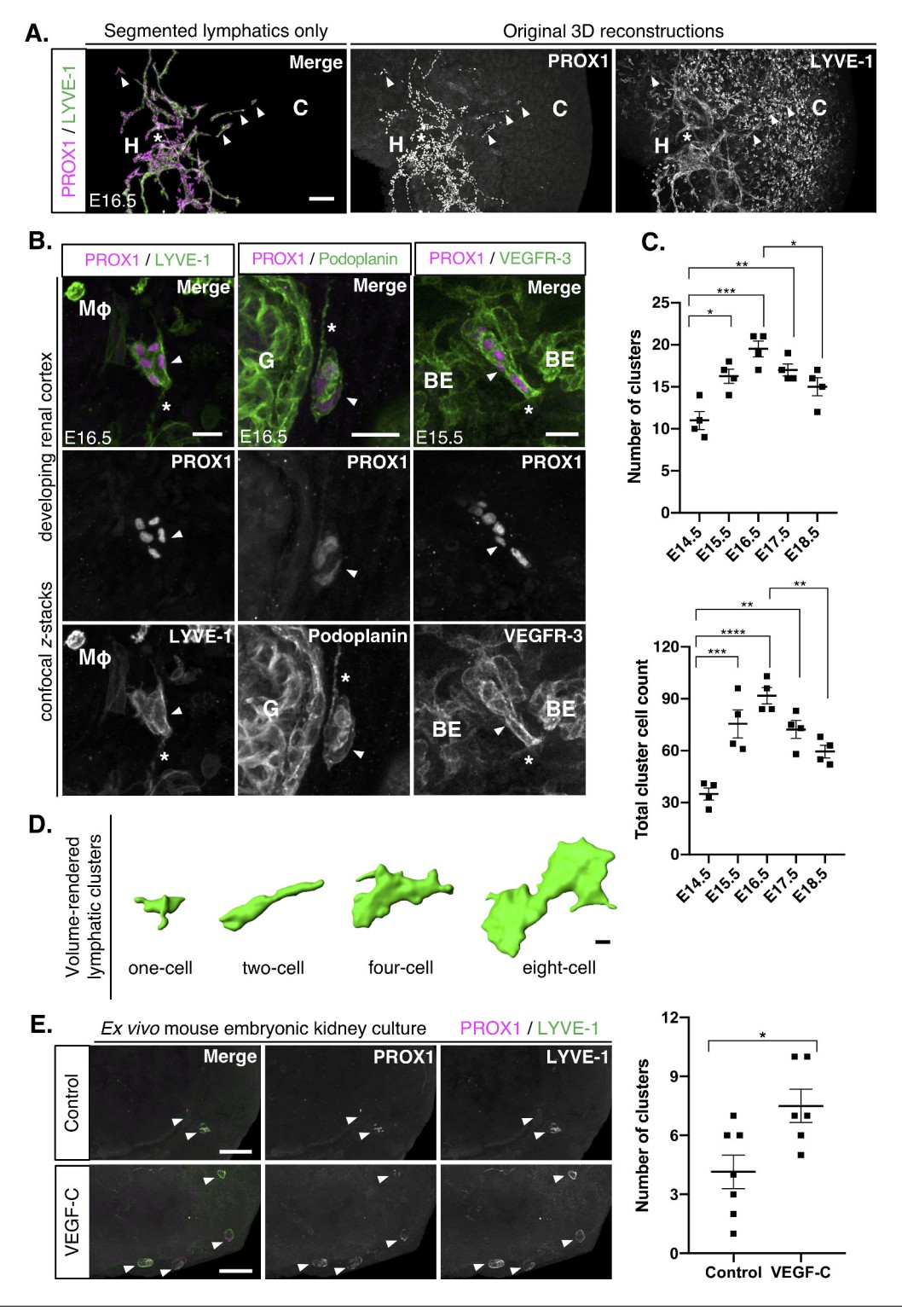

**Figure 2.** Isolated lymphatic endothelial cell clusters in the mouse embryonic kidney that form in response to VEGF-C. (A) In the developing kidney, PROX1+/LYVE-1+ lymphatic endothelial cells appear as either vessels or cell clusters. A representative image from analysis of four kidneys from three independent litters. Lymphatic clusters (white arrowheads) at E16.5 are structurally and anatomically distinguishable from the lymphatic plexus (white asterisk). The segmentation of PROX1+/LYVE-1+ lymphatics makes clusters easier to visualize. Scalebar: 50 µm. H, hilum; C, cortex. (B) Confirmation of the lymphatic identity of clusters. Representative confocal z-stacks

*Figure 2 continued on next page*

*Figure 2 continued*

from a minimum of 3 kidneys from three independent litters stained for PROX1, LYVE-1, podoplanin and VEGFR-3 identifies lymphatic clusters (white arrowhead). Filopodia-like projections (white asterisks) extend from the clusters. The relationship of clusters to other structures in the kidney is also shown. Scalebars: 20 μm. BE, blood endothelium; G, glomerulus; Mφ, macrophage. (C) Quantitative assessment of lymphatic cluster dynamics during mouse kidney development. For each embryonic kidney immunolabelled for PROX1 and LYVE-1, the number of clusters (top graph) or the total number of cells within all clusters (bottom graph) in any single kidney was quantified. Each data point represents the values for one kidney and error bars represent standard error of the mean. ANOVA showed a significant overall increase in the total number of clusters per kidney ($F = 10.9$; p=0.0002) and the total number of cells constituting all clusters per kidney ($F = 15.9$; p<0.0001). Brackets between timepoints demonstrate significance when multiple comparisons were performed using Bonferroni tests. *: p<0.0332; **: p<0.0021; ***: p<0.0002, ****: p<0.0001. Raw data and results of all multiple comparisons are presented in *Figure 2—source data 1*. (D) Volume renderings of a representative E15.5 kidney conveys the heterogeneity of PROX1$^+$/LYVE-1$^+$ cluster size and morphology. Scalebar: 5 μm. (E) VEGF-C increases the abundance of lymphatic clusters ex vivo. Representative 3D reconstructions of mouse embryonic kidneys, harvested at E14.5, cultured at the air-liquid interface for 48 hr with or without 40 ng/ml recombinant VEGF-C and immunolabelled for PROX1 and LYVE-1. PROX1$^+$/LYVE-1$^+$ lymphatic cell clusters (white arrowheads) were detectable in both treated and untreated explants. The number of PROX1$^+$/LYVE-1$^+$ lymphatic cell clusters per kidney was quantified. Each data point represents the values for one cultured kidney and error bars represent standard error of the mean. An unpaired Student's *t*-test showed that treatment with VEGF-C significantly increased the abundance of PROX1$^+$/LYVE-1$^+$ cell clusters within the explants ($t = 2.77$; p=0.0184). Data were pooled from two independent culture experiments. Raw data and results of statistical tests are presented in *Figure 2—source data 2*.

The online version of this article includes the following video, source data, and figure supplement(s) for figure 2:

**Source data 1.** This spreadsheet contains the numerical raw data and statistics for the quantitative analysis of kidney lymphatic endothelial cell cluster dynamics in vivo, shown in *Figure 2C*.

**Source data 2.** This spreadsheet contains the numerical raw data and statistics comparing the effect of VEGF-C treatment on lymphatic endothelial cell cluster number in mouse embryonic kidney explants, shown in *Figure 2E*.

**Figure supplement 1.** Molecular profile of lymphatic endothelial cell clusters.

**Figure 2—video 1.** The appearance of lymphatic endothelial cell clusters in the developing mouse kidney.

https://elifesciences.org/articles/48183#fig2video1

**Figure 2—video 2.** Kidney lymphatic clusters in mouse are responsive to VEGF-C ex vivo.

https://elifesciences.org/articles/48183#fig2video2

---

of lymphatic vessels during this process has not yet been examined. To image lymphatic vessels during human kidney development, we acquired millimeter-thick slices from human fetal kidneys at 12PCW. This stage marks the end of the first trimester in human gestation, designated the fetal stage of human organogenesis, and the human kidney at this stage is approximately equivalent to the E15.5 mouse kidney (*Lindström et al., 2018c*). We used 3D reconstructions of 12PCW human kidneys to identify PROX1$^+$/podoplanin$^+$ lymphatic vessels, including large patent lymphatic vessels in the renal hilum and extensive networks of vessels in the maturing cortex (*Figure 3A*). Analogous to mouse, we identified a population of PROX1$^+$/podoplanin$^+$ lymphatic endothelial cell clusters in 12PCW human kidneys (*Figure 3B* and *Figure 3—video 1*) that were structurally distinct from bona fide lymphatic vessels. Thus, the existence of lymphatic endothelial cell clusters in the kidney is likely a conserved feature in mammalian development. These clusters, akin to those previously described in the dermis, mesentery, meninges and heart, may represent a tissue-specific population of lymphatic progenitor cells, and our work provides the first demonstration of their existence in a developing human organ.

## Defective renal lymphatic vascular architecture in polycystic kidney disease

We then sought to translate our 3D imaging and analysis strategy to a pathological setting in which kidney lymphatic development might be affected. We examined a mouse model of the most prevalent genetic renal anomaly, polycystic kidney disease (PKD), which features the accumulation of fluid-filled epithelial cysts within the kidney that drive renal inflammation and fibrosis (*Bergmann et al., 2018*). We had several reasons for hypothesizing that kidney lymphatics are altered in PKD. Firstly, the gene responsible for the majority of cases of autosomal dominant (AD)PKD, *Pkd1*, is known to

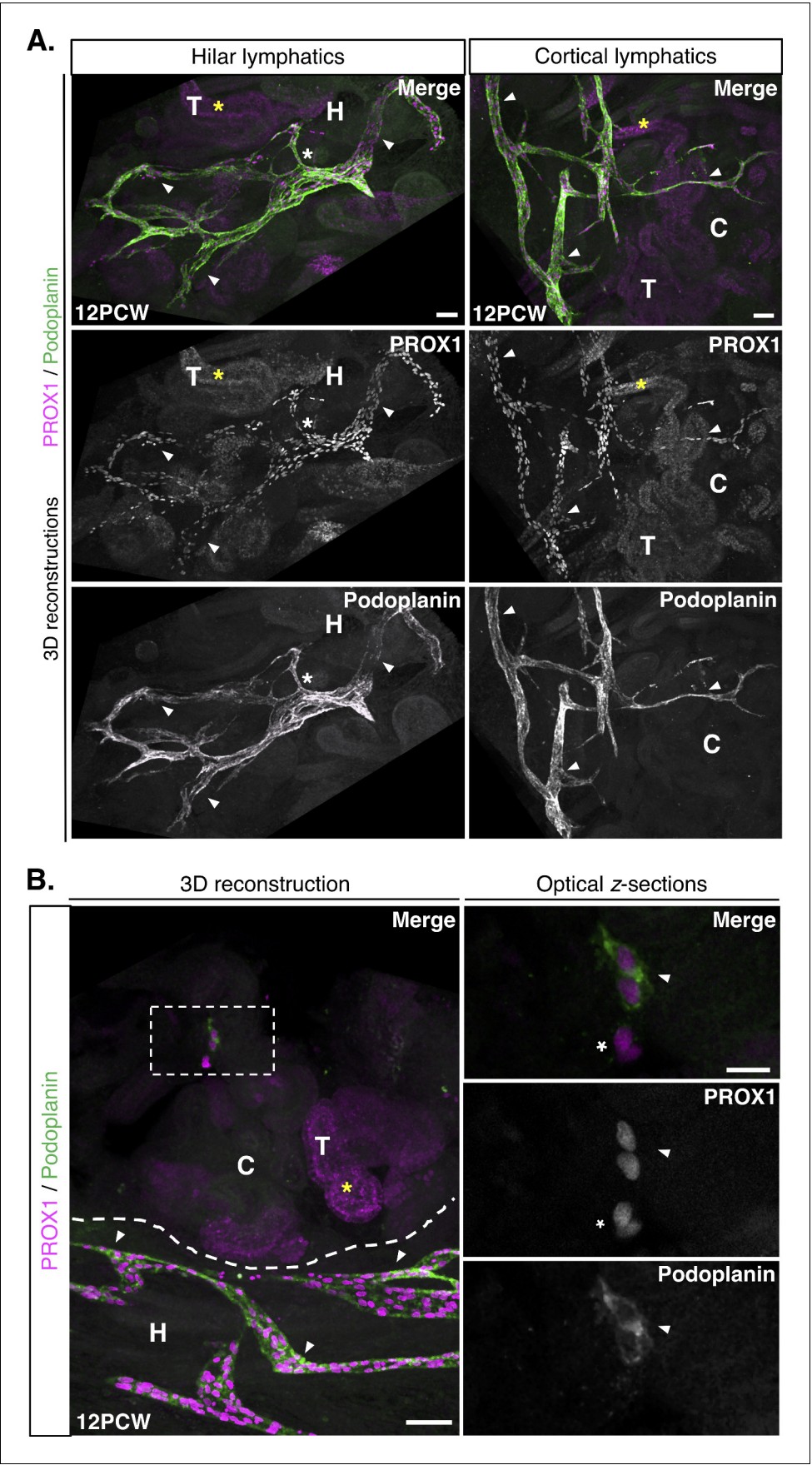

**Figure 3.** Lymphatic vessels and clusters in the 12PCW human fetal kidney. (**A**) Representative 3D reconstructions of 1 mm-thick kidney slices from three independent human fetal kidneys at 12PCW, each stained for PROX1 and podoplanin. We observed PROX1$^+$/podoplanin$^+$ lymphatic vascular networks in the renal hilum and cortex (white arrowheads). Large caliber vessels were visible in the renal hilum (white asterisks). Scalebars: 200 µm. (**B**) Representative 3D reconstructions of junction between the renal hilum and cortex (dashed line) reveals a distinct lymphatic endothelial cell cluster (white boxed region) expressing PROX1 and podoplanin. This was structurally distinct from lymphatic vessels (white arrowheads) in this region of 12PCW kidneys and are shown in higher magnification in the optical *z*-sections (right three panels). Adjacent to the cluster (white arrowhead in optical *z*-section) were PROX1$^+$/podoplanin$^-$ cells of unknown identity (white asterisk). Scalebar: 200 µm (3D reconstruction); 20 µm (optical *z*-section). PROX1 expression was also found in tubular epithelial cells at 12PCW (yellow asterisks throughout). C, cortex; H, hilum; T, tubular epithelium.

The online version of this article includes the following video for figure 3:

**Figure 3—video 1.** Existence of lymphatic vessels and clusters in the human fetal kidney.

https://elifesciences.org/articles/48183#fig3video1

---

regulate extra-renal lymphatic endothelial cell migration and lymphatic vessel morphogenesis, as edema and structural defects in lymphatic vasculature have been observed in *Pkd1* mutant zebrafish and mice (*Coxam et al., 2014*; *Outeda et al., 2014*). Secondly, the endothelial knockout of *Pkd1* in mice decreased branching and increased diameter of dermal lymphatic vessels (*Lindström et al., 2018a*). Finally, delivery of recombinant VEGF-C increased the abundance of lymphatics within the kidney and reduced disease severity in two mouse models of PKD (*Huang et al., 2016*). However, despite a putative link to PKD, the phenotype of lymphatics in polycystic kidneys and their relationship to cysts have not yet been examined.

We assessed the lymphatic vasculature in kidneys from mouse embryos carrying a *Pkd1* p.R3277C allele (*Pkd1$^{RC}$*). This incompletely penetrant allele in homozygosity causes adult onset ADPKD (*Hopp et al., 2012*; *Rossetti et al., 2009*). In homozygous mice (*Pkd1$^{RC/RC}$*), the slow growth of kidney cysts is considered to mimic the temporal progression of human ADPKD compared with other more rapidly progressive mouse models (*Happé and Peters, 2014*). Similar to knockouts of *Pkd1* (*Coxam et al., 2014*) or mice lacking key genes involved in lymphatic development (*Wigle and Oliver, 1999*; *Karkkainen et al., 2004*; *Bos et al., 2011*; *François et al., 2008*), we found that *Pkd1$^{RC/RC}$* homozygous embryos presented with subcutaneous edema at E15.5 (*Figure 4—figure supplement 1*). Using conventional histology (*Figure 4A*) we discerned corticomedullary cysts lined by tubular epithelial cells in E18.5 *Pkd1$^{RC/RC}$* kidneys but not in wildtype littermate controls (*Pkd1$^{+/+}$*), consistent with early stages of cyst formation. These cysts were in close proximity to cortical lymphatic vessels (*Figure 4—video 1*). The proximity between lymphatics and cysts suggests that excess tissue fluid in PKD may be able to enter lymphatic vessels *via* paracellular transport (*Triacca et al., 2017*) and that there might be direct or indirect molecular interactions between cyst epithelium and lymphatic endothelial cells (*Huang et al., 2016*).

To visualize the renal lymphatics in the *Pkd1$^{RC}$* model, we generated 3D visual models of PROX1$^+$/LYVE-1$^+$ segmented lymphatic networks and observed a stunted appearance of lymphatics in *Pkd1$^{RC/RC}$* compared with *Pkd1$^{+/+}$* kidneys at E18.5 (*Figure 4B*). Interestingly, we found that the lymphatic network of E18.5 *Pkd1$^{+/+}$* kidneys, raised on a C57Bl/6 background, was more extensive than that of CD-1 mice assessed at the same embryonic day (see *Figure 1B*). Differences between mouse strains have also been observed when analyzing the incidence of subcutaneous edema, hemorrhage and embryonic lethality upon knockout of key transcription factors or micro-RNAs governing lymphatic development (*François et al., 2008*; *Kontarakis et al., 2018*).

To screen for more subtle lymphatic abnormalities in the kidneys of *Pkd1$^{RC/RC}$* mice, we used filament tracing software to generate color-coded models of vessel diameter at E18.5 (*Figure 4C* and *Figure 4—video 2*). We found large caliber vessels (diameter >70 µm), present at the base of the lymphatic plexus in *Pkd1$^{+/+}$* kidneys, constituting the ring-like anastomosis in the renal hilum. There were no lymphatic vessels of this large diameter in the equivalent region within *Pkd1$^{RC/RC}$* kidneys at E18.5. Quantitative analysis confirmed this phenotype (*Figure 4D*), finding a significant reduction by 18% in the mean diameter of the largest lymphatic vessel in *Pkd1$^{RC/RC}$* kidneys compared to *Pkd1$^{+/+}$* controls (p=0.0087). In contrast, no significant differences were observed in the total number of vessel branches, the range of vessel branch lengths, the total volume of the lymphatic network between

homozygous mutants and wildtype controls (*Figure 4—figure supplement 2*). Although the kidney volume was not significantly different between the two groups, we derived that the number of lymphatic vessel branches per unit kidney volume (p=0.0418) and the proportion of kidney volume occupied by lymphatics (p=0.0367) were significantly reduced in *Pkd1^RC/RC^* kidneys compared to *Pkd1^+/+^* controls (*Figure 4E*). The complex global defect in kidney lymphatic vessel development in the *Pkd1^RC^* model, together with the close cyst-lymphatic relationship, suggests that inadequate clearance of tissue fluid by lymphatic vessels may contribute to the expansion of cysts in PKD.

## Conclusion

In summary, we have combined advanced 3D imaging with structural rendering to provide novel spatial, temporal and quantitative insights into the formation of lymphatic vessels in developing mouse and human kidneys. By creating 3D images of kidney lymphatic vessels at single-cell resolution, we were able to visually and quantitatively demonstrate a period of quiescence, followed by extensive remodeling and expansion of the lymphatic plexus during renal development in mouse, in a process that is likely promoted by VEGF-C. The timing of plexus expansion in mouse kidneys coincides with the appearance of a highly dynamic population of lymphatic endothelial cell clusters, structurally distinct from bona fide lymphatic vessels. These clusters responded to VEGF-C and were conserved in human fetal kidneys. Our imaging approach further revealed defects in lymphatic architecture in polycystic kidneys. Together, we present a comprehensive study of the lymphatic vasculature in the developing mammalian kidney, provide the first evidence that lymphvasculogenesis is conserved in humans and implicate lymphatic anomalies in the pathophysiology of PKD.

# Materials and methods

### Key resources table

| Reagent type | Designation | Source (publication) | Identifier | Additional information |
|---|---|---|---|---|
| Biological sample (*Mus musculus*) | CD-1 wildtype mouse | Charles River | Crl:CD1(ICR) | Maintained as a random bred closed colony in house |
| Biological sample (*Mus musculus*) | *Vegfc^LacZ^* mouse | See *Karkkainen et al. (2004)* | Vegfc^tm1Ali^ RRID:IMSR_EM:10734 | *Vegfc^LacZ/+^* males mated with *Vegfc^+/+^* females to generate experimental litters, maintained on a CD-1 background |
| Biological sample (*Mus musculus*) | *Pkd1* p.R3277C mouse | See *Hopp et al. (2012)* | Pkd1^tm1.1Pcha^ RRID: MGI:5476836 | *Pkd1^RC/+^* animals mated to generate experimental litters, maintained on a C57BL/6 background |
| Biological sample (*Homo sapiens*) | Human fetal kidney | Human Developmental Biology Resource (*Gerrelli et al., 2015*) | - | Material obtained at 12PCW |
| Peptide, recombinant protein | Vascular endothelial growth factor C | R and D Systems | Cat# 9199-VC-025 | Added to culture medium at 40 ng/ml |
| Sequence-based reagent | *Pkd1^RC^*_F | See *Hopp et al. (2012)* | PCR primers | CAAAGGTCTGGGTGATAACTGGTG |
| Sequence-based reagent | *Pkd1^RC^*_R | See *Hopp et al. (2012)* | PCR primers | CAGGACAGCCAAATAGACAGGG |

*Continued on next page*

*Continued*

| Reagent type | Designation | Source (publication) | Identifier | Additional information |
|---|---|---|---|---|
| Antibody | Anti-PROX1 (Goat polyclonal) | R and D Systems | Cat# AF2727 RRID:AB_2170716 | Wholemount immunofluorescence (1:200) in mouse and human. Transcription factor labelling nuclei of lymphatic endothelium (*Wigle and Oliver, 1999*) and some tubular epithelium within the kidney (*Kim et al., 2015*) |
| Antibody | Anti-PROX1 (Rabbit polyclonal) | Millipore | Cat# ABN278 RRID:AB_2811075 | Wholemount immunofluorescence (1:200) in mouse only. |
| Antibody | Anti-LYVE-1 (rabbit polyclonal) | Abcam | Cat# ab14917 RRID:AB_301509 | Wholemount immunofluorescence (1:100) in mouse only. Labels lymphatic endothelial cell surface (*Banerji et al., 1999*). Present on F4/80$^+$ macrophages and venous endothelium in the developing mouse kidney (*Lee et al., 2011*) |
| Antibody | Anti-LYVE-1 (goat polyclonal) | R and D Systems | Cat# AF2125 RRID:AB_2297188 | Colorimetric immunohistochemistry (1:20) of *Vegfc$^{LacZ/+}$* mouse kidney sections. |
| Antibody | Anti-podoplanin (Syrian hamster monoclonal) | ThermoFisher Scientific | Cat# 14-5381-82 Clone: 8.1.1 RRID:AB_1210505 | Wholemount immunofluorescence (1:100) in mouse only. Expressed on lymphatic endothelial cell membranes (*Wetterwald et al., 1996*; *Breiteneder-Geleff et al., 1999*). Also expressed on podocytes within the developing and mature kidney (*Breiteneder-Geleff et al., 1997*) |
| Antibody | Anti-podoplanin (Mouse monoclonal) | Agilent | Cat# M361929-2 Clone: D2-40 RRID:AB_2162081 | Wholemount immunofluorescence (1:100) in human only. Expressed on lymphatic endothelial cell membranes within the human fetal kidney (*Jin et al., 2010*). |
| Antibody | Anti-VEGFR-3 (Goat polyclonal) | R and D Systems | Cat# AF743 RRID:AB_355563 | Wholemount immunofluorescence (1:100) in mouse only. Expressed on lymphatic endothelial cell membranes and blood endothelia in the developing kidney (*Kenig-Kozlovsky et al., 2018*; *Kaipainen et al., 1995*). |
| Antibody | Anti-endomucin (Rat monoclonal) | Santa-Cruz | Cat# sc-53941 Clone: V.5C7 RRID:AB_2100038 | Wholemount immunofluorescence (1:50) in mouse only. |

*Continued on next page*

*Continued*

| Reagent type | Designation | Source (publication) | Identifier | Additional information |
|---|---|---|---|---|
| Antibody | Anti-PECAM-1 (Rat monoclonal) | BD Biosciences | Cat# 553370 Clone: MEC13.3 RRID:AB_394816 | Wholemount immunofluorescence (1:50) in mouse only. |
| Antibody | Anti-F4/80 (Rat monoclonal) | Bio-Rad | Cat# MCA497 Clone: Cl:A3.1 RRID:AB_2098196 | Wholemount immunofluorescence (1:50) in mouse only. |
| Antibody | Anti-UPK3A (Rabbit polyclonal) | Abcam | Cat# ab82173 RRID:AB_2213493 | Wholemount immunofluorescence (1:100) in mouse only. |
| Antibody | Anti-AQP2 (Rabbit polyclonal) | Abcam | Cat# ab15116 RRID:AB_301662 | Wholemount immunofluorescence (1:100) in mouse only. |
| Other | Hoechst 33342 | Life Technologies | Cat# H3570 | Nuclear counterstain. Wholemount immunofluorescence (1:2000) |
| Software, algorithm | ImageJ/FIJI | https://imagej.net/Welcome (*Schindelin et al., 2012*) | RRID:SCR_002285 | Version 2.0 |
| Software, algorithm | Zen | https://www.zeiss.com /microscopy/ int/products/microscope -software/zen.html | - | - |
| Software, algorithm | Imaris | http://www.bitplane.com | RRID:SCR_007370 | Version 8.2. Imaris FilamentTracer used in this paper (RRID:SCR_007366) |
| Software, algorithm | Amira | https://www.thermofisher.com/ uk/en/home/industrial/electron- microscopy/electron-microscopy- instruments-workflow -solutions/3d-visualization- analysis-software/amira-life -sciences-biomedical.html | RRID:SCR_007353 | Version 5.4 |

## Acquisition of mouse embryonic and human fetal kidneys

All experiments were carried out according to a UK Home Office project license (PPL: PE52D8C09) and were compliant with the UK Animals (Scientific Procedures) Act 1986. To assess lymphatic development in wildtype mouse kidneys, we collected embryos from outbred CD-1 mice; maintained as a random bred closed colony at our facility. To determine the expression of VEGF-C during kidney development, we mated *Vegfc^{LacZ/+}* males (*Karkkainen et al., 2004*), backcrossed to a CD-1 background, with CD-1 wildtype females and collected embryos at desired timepoints. For analysis of lymphatic development in a mouse model of PKD, we performed matings between mice heterozygous for the *Pkd1^{RC}* mutation (*Hopp et al., 2012*) on a C57BL/6 background to generate experimental litters. For all mouse experiments, matings were set up in the evening and copulation plugs found the following morning at 9am were designated E0.5. At the desired timepoint, pregnant dams were sacrificed by $CO_2$ inhalation and death confirmed by cervical dislocation. Mouse embryos were excised from the uterus and dissected in 1 x phosphate-buffered saline (PBS). Tails were acquired and stored at $-20°C$ from embryos obtained from pregnant *Pkd1^{RC/+}* heterozygous females for genotyping, which was performed as previously described (*Hopp et al., 2012*). At this point, embryos generated from *Pkd1^{RC/+}* crosses were imaged using a Zeiss Axio Lumar V12 stereomicroscope. Human fetal kidneys were obtained from the Human Developmental Biology Resource (http://www.hdbr.org), which obtains written consent from donors to collect, store and distribute human fetal material between 4-20PCW (*Gerrelli et al., 2015*). Following dissection, human fetal material was maintained in tissue culture medium at 4°C. With the exception of *Vegfc^{LacZ/+}* kidneys, we immediately washed all biological material in washed in 1 x PBS to remove blood, fixed the tissues overnight in 4% (w/v) paraformaldehyde (PFA) in PBS at 4°C, and washed tissues twice in PBS the next day. Human fetal kidneys were then incubated in 30% (w/v) sucrose in PBS overnight at 4°C,

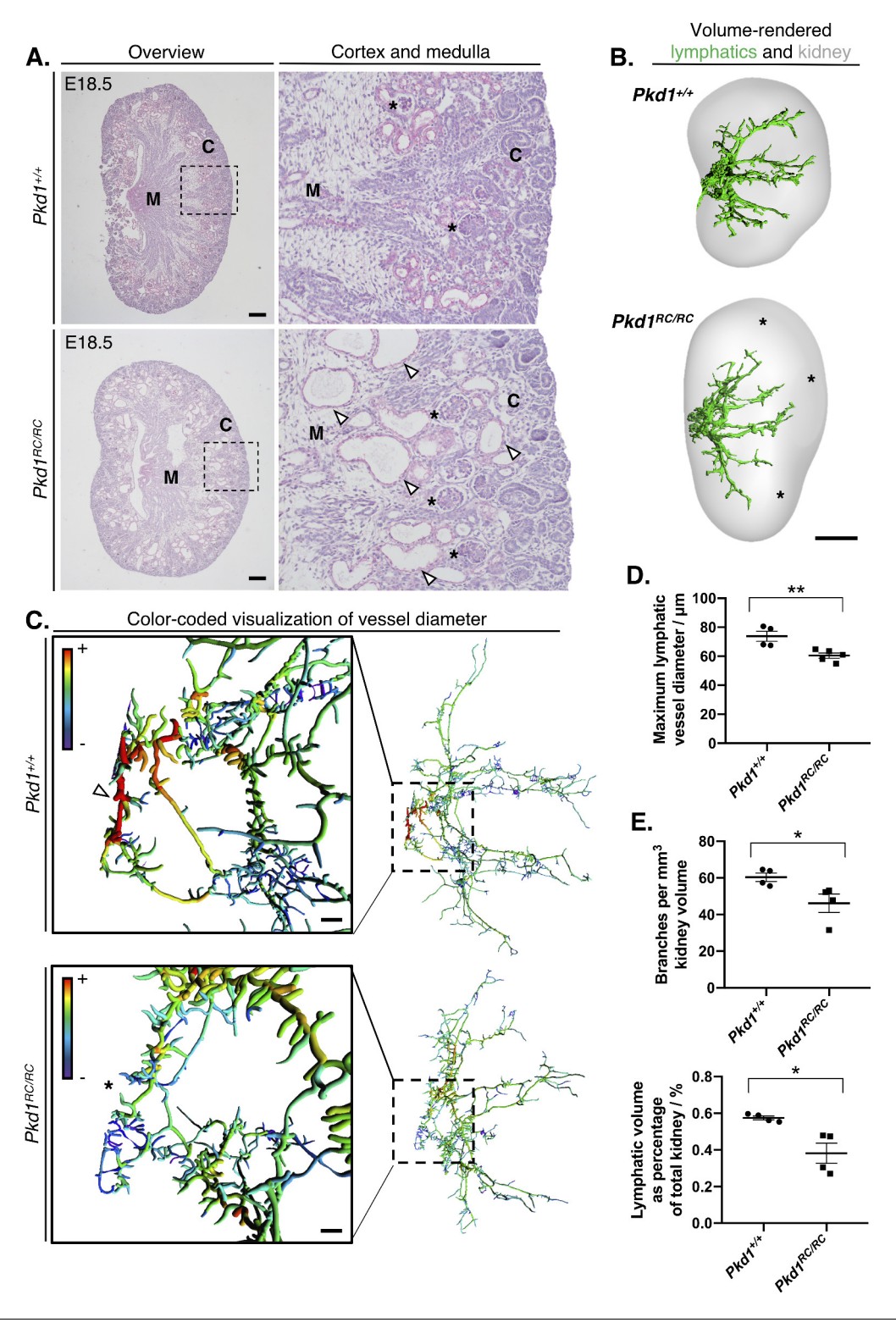

**Figure 4.** Identification of kidney lymphatic vessel defects in the *Pkd1^RC* mouse model. (**A**) Representative brightfield images of four histology slides each from right kidneys of three independent *Pkd1^RC/RC* or *Pkd1^+/+* mouse embryos at E18.5. We cut 5 μm sections cut from the middle of *Pkd1^RC/RC* and *Pkd1^+/+* kidneys and stained with periodic acid-Schiff. Panels on the right show magnified images of the boundary between the cortex and medulla in either group (boxed region in low magnification images), where corticomedullary cysts (white

*Figure 4 continued on next page*

*Figure 4 continued*

arrowheads) are visible in $Pkd1^{RC/RC}$ kidneys at this timepoint. Glomeruli in either group are shown with black asterisks. Scalebars: 200 µm. C, cortex; M, medulla. (B) Volume rendered kidneys and lymphatic vessels in $Pkd1^{RC/RC}$ and $Pkd1^{+/+}$ kidneys. Overall, the lymphatic network appeared stunted in homozygous mutants and regions within the peripheries of their kidneys were unoccupied by lymphatic vessels at this stage (see black asterisks). Scalebars: 500 µm. (C) Filament tracing and color-coded visualization reveals a structural defect in large-caliber lymphatic vessel diameter in $Pkd1^{RC/RC}$ mice at E18.5. Each vessel branch is color-coded on a spectrum, from the smallest diameter vessels (average branch diameter <20 µm) colored in purple to the largest vessel branch (average branch diameter >70 µm) colored in red. Insets on the left showed the magnified region of the plexus on the right, corresponding to the ring-like anastomosis at the base of the plexus. The large diameter red colored vessel branches in the $Pkd1^{+/+}$ kidneys (white arrowhead) are absent in the equivalent region in $Pkd1^{RC/RC}$ kidneys (black asterisk). Scalebars: 50 µm. (D) Comparison of maximum vessel diameter in $Pkd1^{+/+}$ and $Pkd1^{RC/RC}$ kidneys at E18.5. An unpaired Student's $t$-test showed that the maximum lymphatic vessel diameter was significantly reduced in $Pkd1^{RC/RC}$ kidneys ($t = 3.6$ p=0.0087). (E) Lymphatic vessel branches and total lymphatic network volume are reduced per unit of kidney volume in $Pkd1^{RC/RC}$ kidneys. ANOVA showed that both the number of vessel branches per mm of kidney ($t = 2.6$, p=0.0418) and the percentage occupancy of the volume of the kidney by lymphatic vessels (Welch's corrected $t = 3.4$, p=0.0367) were significantly reduced in homozygous mutants compared to wildtype controls at this timepoint. Welch's correct was applied when testing percentage occupancy of the total kidney volume by lymphatic vessels, as the two groups did not have equality of variances for this parameter. All kidneys from this analysis were pooled from a total of three independent litters. In all graphs, each data point represents the right kidney from an individual embryo and error bars represent standard error of the mean. Raw data and results of all statistical tests are presented in *Figure 4—source data 1*.

The online version of this article includes the following video, source data, and figure supplement(s) for figure 4:

**Source data 1.** This spreadsheet contains the numerical raw data and statistics for the quantitative analysis of kidney lymphatic vessel phenotypes in the $Pkd1^{RC}$ mouse model at E18.5 shown in *Figure 4D*, *Figure 4E* and *Figure 4—figure supplement 1*.

**Figure supplement 1.** Mouse embryos homozygous for the $Pkd1^{RC}$ allele present with subcutaneous edema and hemorrhage.

**Figure supplement 2.** Further quantitative analysis of lymphatic defects in the $Pkd1^{RC}$ mouse model.

**Figure 4—video 1.** Close proximity between lymphatic vessels and renal cysts in the $Pkd1^{RC}$ mouse model.
https://elifesciences.org/articles/48183#fig4video1

**Figure 4—video 2.** Reduced diameter of large-caliber lymphatic vessels in the $Pkd1^{RC}$ mouse model.
https://elifesciences.org/articles/48183#fig4video2

---

prior to embedding in 4% (w/v) agarose in PBS and slicing at 1 mm on a vibrating blade microtome (VT1000S, Leica Biosystems). We subsequently stored all material in PBS with 0.02% (w/v) sodium azide at 4°C.

## Wholemount X-gal staining and imaging

$Vegfc^{LacZ/+}$ or $Vegfc^{+/+}$ kidneys were fixed in 2% PFA for 20 min on ice. They were then washed thrice for five mins each in PBS, before incubating at 37°C in stain solution (5 mM potassium ferricyanide, 5 mM potassium hexacyanoferrate (III) trihydrate, 2 mM magnesium chloride and 0.01% (w/v) sodium deoxycholate in PBS) with X-gal in a 1:40 ratio. The X-gal reaction was quenched by washing twice in PBS with 0.02% sodium azide. Wholemount X-gal images were captured using a Leica MZFLIII stereomicroscope with an IDS UI-3080CP-C-HQ camera and micromanager software.

## Histology, colorimetric immunohistochemistry and brightfield imaging

$Pkd1^{RC/RC}$, $Pkd1^{+/+}$ or X-gal-stained $Vegfc^{LacZ/+}$ kidneys were dehydrated in an ethanol series prior to incubation in Histo-Clear II (National Diagnostics) and embedding in paraffin wax. Sections were cut at 5 µm using a microtome. Deparaffinization and staining with periodic acid-Schiff was performed as previously described (*Brzóska et al., 2016*). Colorimetric immunostaining of $Vegfc^{LacZ/+}$ kidney sections for LYVE-1 was performed using a goat anti-mouse LYVE-1 with horseradish peroxidase rabbit anti-goat secondary antibody. Brightfield imaging was performed using an upright Zeiss Axioplan microscope equipped with a Zeiss Axiocam HRc camera and Axiovision software.

## Ex vivo culture of mouse embryonic kidneys

Freshly isolated kidneys were obtained from E14.5 CD-1 mice, explanted on Millicell cell culture inserts (Millipore) and incubated at 37°C in 20% $O_2$ and 5% $CO_2$. Explants were grown at the air-liquid interface with Dulbecco's Modified Eagle Medium/Nutrient Mixture F-12 (Gibco), supplemented with or without 40 ng/ml recombinant VEGF-C. The culture medium was replaced 24 hr after initiation of the experiment. After 48 hr in culture, explants were collected in PBS, fixed in 4% PFA overnight and processed for wholemount immunolabelling, optical clearing and confocal microscopy (see below). Infected explants and those that had not adhered to the cell culture membrane were discarded. Data were pooled from two independent experiments, each from a separate litter of CD-1 embryos.

## Wholemount immunofluorescence and optical clearing

For mouse embryonic kidneys, a modified version of the iDISCO pipeline (*Renier et al., 2014*) was performed for wholemount indirect immunolabelling using antibodies and concentrations indicated in the **Key resources table,** with all steps performed on rotating shakers (*Jafree et al., 2020*). We first dehydrated mouse embryonic kidneys in a methanol series at room temperature (RT) and bleached in methanol with 5% (v/v) of 30% hydrogen peroxide ($H_2O_2$) solution overnight at 4°C. They were then rehydrated at RT and permeabilized in PBS with 0.2% (v/v) Triton X-100, 2.3% (w/v) glycine and 20% (v/v) dimethyl sulfoxide (DMSO) overnight at 4°C. Blocking was then performed in PBS with 0.2% Triton X-100, 6% (v/v) donkey serum and 10% DMSO at RT for one day, and incubated in 500 µl of wash buffer (PBS solution with 0.2% (v/v) Tween-20, 0.1% (v/v) of 10 mg/ml heparin stock solution) with 3% donkey serum, 5% DMSO and primary antibody for three days at 4°C. The labelled kidneys were then incubated in washing buffer 4–6 times for 1 hr each at RT. Secondary antibodies, either AlexaFluor 488, 546 or 633 (ThermoFisher Scientific) were then added at a concentration of 1:200 in 500 µl of wash buffer with 3% donkey serum and 5% DMSO and incubated at 4°C overnight, before 4–6 applications of washing buffer alone for 1 hr each. Hoechst was added with secondary antibodies for selected *Pkd1^{RC/RC}* or *Pkd1^{+/+}* kidneys.

The wholemount immunofluorescence process was similar for millimeter-thick vibratome slices of 12PCW human kidneys, but instead adapting a protocol optimized for human embryonic and fetal material (*Belle et al., 2017*). The protocol was identical to above, but involved the following changes: (1) bleaching was performed in 10% of 30% $H_2O_2$ solution; (2) permeabilization was performed at RT in 1 x PBS with 0.2% (w/v) bovine gelatin with 0.2% Triton X-100 (PBSGT) with 20% DMSO overnight; (3) all wash steps were performed in PBSGT; (4) Blocking was performed in PBSGT with 10% DMSO and 3% donkey serum and (5) Primary and secondary antibody incubations were performed in 1 ml of PBSGT with 0.1% (w/v) saponin, 5% DMSO and 3% donkey serum at 37°C for six days, and primary antibodies were replenished after three days. After immunolabelling, both mouse embryonic kidneys and human kidney slices were dehydrated in a methanol series at RT, adapted for clearing using a 1:1 mixture of methanol and 1:2 benzyl alcohol:benzyl benzoate, termed BABB (*Combes et al., 2014*; *Dodt et al., 2007*), and finally cleared in BABB alone until optically transparent.

## Confocal imaging with single- or two-photon excitation

All confocal images were acquired using a Zeiss LSM 880 Upright Confocal Multiphoton microscope with gallium arsenide phosphide internal and external detectors and a 10x/numerical aperture 0.5 water dipping objective (working distance: 3.7 mm) with 2.77 µm z-step. To protect the microscope objective from BABB, we suspended smaller tissues (E14.5–16.5 mouse kidneys) in a drop of BABB within a glass-bottomed FluoroDish (World Precision Instruments), which was inverted for upright confocal imaging. Larger tissues (E17.5–18.5 mouse kidneys and 12PCW human kidney slices) were placed in BABB within a custom imaging chamber, consisting of an FKM rubber ring (Polymax) between a glass slide and coverslip. Prior to imaging, a drop of distilled water was placed on top of the inverted FluoroDish or coverslip, into which the objective was placed. For single-photon excitation we used laser lines at 488, 561 and 633 nm. For two-photon excitation, a Mai Tai eHP DeepSee multiphoton laser (SpectraPhysics, 800 nm) was used. Confocal z-stacks were acquired as 8-bit images with pixel resolutions of 512 × 512 or 1024 × 1024. For E16.5–18.5 mouse embryonic

kidneys and 12PCW human fetal kidney slices, tile scanning was performed with a 10% overlap between each image tile. All images were saved as CZI files.

## Image processing and analysis

We use the Stitching tool in Zen software (Zeiss) to stitch together tile scans acquired from confocal imaging. In kidneys from the $Pkd1^{RC}$ mouse model stained with Hoechst or in highly autofluorescence 12PCW human kidney slices, spectral unmixing was performed in Zen to separate nuclear counterstaining and autofluorescence respectively from spill-over into other fluorescence channels. Images were then exported to FIJI. Confocal image stacks were separated into individual fluorescence channels and Despeckle and Sharpen tools were used to reduce non-specific background fluorescence. Where maximum intensity projections or optical z-sections were required, scalebars applied and exported as TIFF files. TIFF files were further imported into Imaris (Bitplane) where 3D reconstruction, segmentation or analysis, volume rendering or videos were required.

## Segmentation and volume rendering

To segment lymphatic vessel and clusters from mouse embryonic kidneys, we used the Isosurface Rendering tool in Imaris. Segmentation of the LYVE-1 channel was performed based on fluorescence intensity, and thresholds were manually selected to capture PROX1$^+$/LYVE-1$^+$ structures. To isolate the lymphatic vessel plexus alone, a filter was applied to retain structures with the largest volume. Thereafter, we generated masks selecting for expression of both PROX1 and LYVE-1 only, generating new channels with either unedited fluorescence intensity (as in *Figure 2A* and *Figure 4—video 1*) or binarized for filament tracing analysis. We volume-rendered and segmented lymphatic vessels and clusters in Imaris to generate the 3D models shown in *Figure 1B* and *Figure 4B*. Due to the high sensitivity of the two-photon detection system, we generated a custom script in FIJI to create a new fluorescence channel containing segmented and binarized PROX1$^+$/LYVE-1$^+$ lymphatic vessels from $Pkd1^{RC/RC}$ or $Pkd1^{+/+}$ kidneys. Briefly, PROX1$^+$ nuclei and LYVE-1$^+$ cells were separately segmented to form masks using a rolling ball background subtraction, auto-thresholding and the 3D Simple Segmentation plugin. The nuclear and cellular masks were overlaid to create a new image channel containing only PROX1$^+$/LYVE-1$^+$ cells. These were then exported into IMARIS as TIFF files alongside the original channels and rendered using Imaris as above.

## Filament tracing and quantification kidney lymphatic vascular development

Color-coded 3D visualization of vascular parameters (as in *Figure 1—figure supplement 4* or *Figure 4C*), such as vessel volume or diameter were generated using the FilamentTracer tool in Imaris from segmented PROX1$^+$/LYVE-1$^+$ lymphatic vessels. For quantitative analysis, TIFF image stacks were imported into Amira (ThermoFisher Scientific). The Filament Editor tool was used in Amira to generate spatial statistical parameters including vessel branch number, lengths, diameters and volumes from each segmented lymphatic plexus. These were exported as CSV files for graphing and statistical tests.

## Quantification of lymphatic endothelial cell cluster frequency and total number of cluster cells

We defined clusters as PROX1$^+$/LYVE-1$^+$ cells with no continuity of PROX1$^+$ nuclei with the lymphatic vessel plexus. To maximize specificity, clusters were quantified manually by rigorously scrolling through serial confocal images of entire mouse embryonic kidneys in FIJI. Where a cluster was observed, it was marked with the Multi-point tool to avoid counting the same clusters twice, and the number of cells within clusters was defined by the number of PROX1$^+$ nuclei they contained. Where tile-scanning was performed, individual tiles were each inspected for the presence of clusters.

## Estimation of total kidney volume, branches per unit kidney volume and proportion of the kidney occupied by lymphatic vessels

Total kidney volume was determined in a semi-automated fashion using FIJI. Series confocal images from each kidney had a Gaussian filter applied ($\sigma$ = 5). Thresholding was applied using the Li method. A binary mask was then created, and holes filled before 3D Connected Component

Analysis. The branches per unit volume of the kidney and total kidney volume occupied by lymphatic vessels were calculated by dividing the total number of vessel branches and the total lymphatic network volume of each kidney, derived from Filament Editor tool in Amira, by the total kidney volume as calculated above. Lymphatic vessel branches per unit kidney volume were expressed per mm, whereas the proportion of the kidney occupied by lymphatic vessels was expressed as a percentage of total kidney volume.

### Sample size estimation

We estimated sample sizes based on preliminary experiments and prior quantitative analyses of developing cardiac (*Klotz et al., 2015*), dermal (*Pichol-Thievend et al., 2018*) and mesenteric (*Stanczuk et al., 2015*) lymphatic vessels. For quantitative analysis of lymphatic development in wild-type mice, we predicted between 4–6 embryonic kidneys per timepoint would be sufficient to power experiments. For all experiments on wildtype mouse embryonic kidneys, 4–6 experimental repeats were randomly selected from kidneys pooled by litter at the required timepoint. In all cases, conclusions were drawn from a minimum of two to three independent litters per timepoint. For analysis of lymphatic abnormalities in the $Pkd1^{RC}$ model at E18.5, we analyzed the right kidney only and the left was taken for histology. All other sample sizes are shown in the relevant figure legends.

### Statistical analysis and data presentation

Storage, statistics and graphing of numerical data was performed in Prism (v8, GraphPad). A two-tailed p value of less than 0.05 was considered statistically significant for all tests. Shapiro-Wilk and Brown-Forsythe tests were used to test Gaussian distribution and equality of variance respectively within all datasets. Where Gaussian distribution and equality of variances were satisfied, we used ANOVA to compare vascular parameters or cluster number and cell content across all timepoints during renal development. Bonferroni tests for multiple comparisons were used to compare individual timepoints after ANOVA. Similarly, we used unpaired Student's *t*-test to compare vascular parameters in $Pkd1^{RC/RC}$ and $Pkd1^{+/+}$ kidney lymphatic vessels and to compare the effect of VEGF-C upon number of lymphatic clusters in mouse embryonic kidney explants. Where Gaussian distribution or equality of variances were violated, non-parametric tests and subsequent multiple comparisons were performed as detailed in the figure legends. All images were compiled into Microsoft PowerPoint where figures were created. All videos, 3D reconstructions, color-coded models and volume renderings were prepared in IMARIS and exported as MP4 or TIFF files. MP4 files were annotated in iMovie (v10.1, Apple Inc).

## Acknowledgements

We thank Professor Juan Pedro Martinez-Barbera (University College London), Dr Peter Baluk (University of California San Francisco) and Dr Nicolas Renier (L'Institut du Cerveau et de la Moelle Épinière) for help and advice. All mice were maintained by staff at GOSICH Western Laboratories and UCL Biological Services. Microscopy was performed at the Light Microscopy Core Facility at UCL GOSICH. All work was performed with the support of the National Institute for Health Research (NIHR) Biomedical Research Centre at Great Ormond Street Hospital for Children NHS Foundation Trust and University College London.

## Additional information

### Funding

| Funder | Grant reference number | Author |
|---|---|---|
| Great Ormond Street Institute of Child Health | Child Health Research PhD Studentship | Daniyal J Jafree Peter J Scambler David A Long |
| University College London | MB/PhD Studentship | Daniyal J Jafree |
| Medical Research Council | MR/P018629/1 | David A Long |
| Medical Research Council | MR/L002744/1 | Adrian S Woolf |

| | | |
|---|---|---|
| Medical Research Council | MR/K026739/1 | Adrian S Woolf |
| British Heart Foundation | FS/19/14/34170 | Rosa Maria Correra |
| Diabetes UK | 15/0005283 | David A Long |
| National Institute for Health Research | NIHR Great Ormond Street Hospital Biomedical Research Centre Award 17DD08 | Dale Moulding |
| British Heart Foundation | CH/11/1/28798 | Paul R Riley |
| British Heart Foundation | RG/15/14/31880 | Peter Scambler |
| Kidney Research UK | Paed_RP_10_2018 | Daniyal J Jafree Adrian S Woolf David A Long |
| Kidney Research UK | IN_012_2019 | Daniyal J Jafree David A Long |

The funders had no role in study design, data collection and interpretation, or the decision to submit the work for publication.

### Author contributions

Daniyal J Jafree, Conceptualization, Resources, Data curation, Software, Formal analysis, Validation, Investigation, Visualization, Methodology, Project administration; Dale Moulding, Resources, Data curation, Software, Visualization, Methodology; Maria Kolatsi-Joannou, Nuria Perretta Tejedor, Natalie J Milmoe, Rosa Maria Correra, Investigation, Methodology, Approved final version of the manuscript; Karen L Price, Resources, Investigation, Project administration, Approved final version of the manuscript; Claire L Walsh, Resources, Data curation, Software, Formal analysis, Validation, Visualization, Methodology, Approved final version of the manuscript; Paul JD Winyard, Conceptualization, Resources, Approved final version of the manuscript; Peter C Harris, Resources; Christiana Ruhrberg, Conceptualization, Investigation, Writing - review and editing; Simon Walker-Samuel, Resources, Software, Methodology; Paul R Riley, Adrian S Woolf, Conceptualization, Validation, Methodology; Peter J Scambler, David A Long, Conceptualization, Resources, Data curation, Supervision, Funding acquisition, Validation, Investigation, Methodology, Project administration

### Author ORCIDs

Daniyal J Jafree (iD) https://orcid.org/0000-0001-8235-0394
Paul R Riley (iD) http://orcid.org/0000-0002-9862-7332
Adrian S Woolf (iD) http://orcid.org/0000-0001-5541-1358
David A Long (iD) https://orcid.org/0000-0001-6580-3435

### Ethics

Human subjects: Human fetal kidneys were obtained from the Human Developmental Biology Resource (http://www.hdbr.org), which obtains written consent from donors to collect, store and distribute human fetal material between 4-20PCW.
Animal experimentation: All experiments were carried out according to a UK Home Office project license (PPL: PE52D8C09) and were compliant with the UK Animals (Scientific Procedures) Act 1986.

### Decision letter and Author response

Decision letter https://doi.org/10.7554/eLife.48183.sa1
Author response https://doi.org/10.7554/eLife.48183.sa2

## Additional files

### Supplementary files

- Source code 1. FIJI Macro for segmentation of PROX1+/LYVE-1+ structures.
- Transparent reporting form

## Data availability

The FIJI script used for segmenting and binarizing PROX1+/LYVE1+ cells has been provided as Source code file 1. All raw numerical data and results of statistical tests are attached as source data for the appropriate figures.

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
