## [Decision Letter]

**Acceptance summary:**

Using three-dimensional imaging, the authors show that lymphatics first appear half-way through murine kidney development, followed by a rapid phase of growth as the kidney starts filtering blood. They find clusters of lymphatic cells that may contribute to later maturation. Finally, they find that kidney lymphatics are distorted under cystic kidney states well before disease progression is obvious. These are important findings that help elucidate the role of lymphatics in both development and disease.

**Decision letter after peer review:**

Thank you for submitting your article "Spatiotemporal dynamics and heterogeneity of mammalian kidney lymphatics in development and disease" for consideration by *eLife*. Your article has been reviewed by two peer reviewers, and the evaluation has been overseen by a Reviewing Editor and Marianne Bronner as the Senior Editor. The following individuals involved in the review of your submission have agreed to reveal their identity: Dontscho Kerjaschki (Reviewer #2).

Summary:

This study is novel and adds substantially to the field since very little is known about lymphangiogenesis (and lymphvasculogenesis) in kidney development and in disease. It uses state of the art imaging in embryonic kidney samples (including human). It convincingly describes initial steps in the genesis of renal lymphatic vessels in mice and man. The techniques used reveal stunning pictures and clearly illustrate the author's claims. The descriptions of lymphatic cell "clusters" is novel to kidney development.

The title of the paper is somewhat misleading and suggests that the study includes several diseases. Thus, a more specific title is needed.

One main concern is that the paper remains essentially descriptive. The first part of the manuscript describes lymphatic development in kidneys in detail. However, the relation to disease is the weakest part that is difficult to connect with the rest of the study. It is not clear how the phenotype of polycystic kidney is essentially related to the lymphatics.

Essential revisions:

*Pkd1* deletion can be induced in vasculature (Coxam et al., 2014). Any evidence of kidney defects in this model?

Could the authors provide co-localization of lymphatic markers with VEGF-C? It would add considerably to the manuscript if kidney VEGF-C expression was also shown in development in relation to lymphatic development. This would be of particular interest in the *Pkd1^RC/RC^*vs *Pkd1^+/+^* mice and should increase our understanding of why lymphatic vessel development is stunted in polycystic kidney disease. This might be possible on existing tissue collected from mice? Is VEGF-C required for the development of kidney lymphatics?

The authors should add a summary of the use of the different lymphatic markers and what they each may or may not relate to.

The lymphatic vessel diameter, branch points and volume between *Pkd1^+/+^* and CD1 mice at the same time in development (E18.5) seem very different. Even more so than between *Pkd1^RC/RC^*and *Pkd1^+/+^* mice. The authors should explain what is the explanation for these differences. Some discussion about differences in lymphatic development between mouse strains should be added.

[Editors' note: further revisions were requested prior to acceptance, as described below.]

Thank you for resubmitting your work entitled "Spatiotemporal dynamics and heterogeneity of renal lymphatics in mammalian development and cystic kidney disease" for further consideration at *eLife*. Your revised article has been favorably evaluated by Marianne Bronner (Senior Editor) and a Reviewing Editor.

The manuscript has been improved but there are some remaining issues that need to be addressed before acceptance.

In the rebuttal, the authors note: "However, our attempts at localising VEGF-C in kidney development using immunostaining or the*Vegfc^LacZ^* reporter mouse were thus far unsuccessful due to poor quality staining and a failure to obtain time-mated embryos at the required stages, respectively." This analysis should be completed, as it is necessary to support the claim that VEGF-C is responsible for lymphatic vessel growth in the kidney, when it could be VEGF-D instead. VEGF-C needs to be expressed (nearby) to make the claim in the manuscript.

---

## [Author Response]

[…]The title of the paper is somewhat misleading and suggests that the study includes several diseases. Thus, a more specific title is needed.

As suggested, we have changed the title to “Spatiotemporal dynamics and heterogeneity of renal lymphatics during mammalian development and cystic kidney disease” to more accurately reflect the content of the paper.

One main concern is that the paper remains essentially descriptive.

We agree that the original version of the manuscript was largely descriptive. However, given the lack of knowledge regarding lymphangiogenesis in kidney development and disease, we feel that high-quality descriptive studies such as ours are required to provide a basis for functional experiments in the future. As a starting point for functional work on the kidney lymphatics, we have performed an additional set of experiments in the revised paper by examining the potential role of VEGF-C in kidney lymphatic development. To do this, E14.5 kidneys were isolated and cultured ex vivo with or without recombinant VEGF-C. Three-dimensional imaging 48 hours later demonstrated scattered PROX1^+^/LYVE-1^+^ cell clusters in control kidney explants, whose abundance increased significantly in explants supplemented with recombinant VEGF-C. These new functional studies are presented in a revised Figure 2E and Figure 2—video 2 and provide evidence that VEGF-C promotes renal lymphatic development. Please see paragraph three of subsection “Characterization of conserved lymphatic endothelial cell clusters in the developing mammalian kidney” for the description of these new experiments in the main text.

The first part of the manuscript describes lymphatic development in kidneys in detail. However, the relation to disease is the weakest part that is difficult to connect with the rest of the study. It is not clear how the phenotype of polycystic kidney is essentially related to the lymphatics.

We agree that our explanation for examining lymphatics in PKD could have been strengthened. We feel that there are several important strands of evidence to justify this link. Firstly, the gene responsible for the majority of cases of autosomal dominant (AD)PKD, *Pkd1,* is known to regulate lymphatic endothelial cell migration and lymphatic vessel morphogenesis, as edema and structural defects in lymphatic vasculature have been observed in *Pkd1* mutant zebrafish and mice (Outeda et al., 2014; Coxam et al., 2014). To translate these findings into putative link between lymphatics and PKD, we now show edema at E15.5 in *Pkd1^RC/RC^* mice, indicative of lymphatic defects in this model (new Figure 4—figure supplement 1). Secondly, the endothelial knockout of *Pkd1* in mice reduces branching and increases diameter of dermal lymphatic vessels (Coxam et al., 2014). Although the kidney was not examined in these studies, this evidence highlights a potential role for *Pkd1* in lymphangiogenesis. Thirdly, we showed in a prior study that delivery of recombinant VEGF-C both increases the abundance of renal lymphatics and reduces disease severity in two mouse models of PKD. The increase in the abundance of renal lymphatics may help to improve the edema and inflammation seen in these PKD models (Huang et al., 2016). Despite the putative link between lymphatics and PKD, the phenotype of lymphatics in polycystic kidneys and their relationship to cysts have not been examined. The results detailed in our manuscript are the first to identify structural anomalies of kidney lymphatics in a mouse model of PKD. The points raised above have been made in subsection “Defective renal lymphatic vascular architecture in polycystic kidney disease” of the revised manuscript to provide a stronger rationale for examining lymphatics in PKD.

Essential revisions:Pkd1 deletion can be induced in vasculature (Coxam et al., 2014). Any evidence of kidney defects in this model?

*Pkd1* deletion has been performed in endothelium in mice using inducible *Sox18-Cre* and constitutive *Tie2-Cre* lines. The investigators showed that these mice had fewer lymphatic branch points and increased average width in the lymphatics of the skin (Coxam et al., 2014). We have highlighted this paper in more detail in the revised manuscript. Unfortunately, the kidneys were not examined in this study. To provide definitive evidence for a kidney lymphatic defect in mice lacking *Pkd1* in the vasculature would require use of multiple *Cre* lines in parallel and other experimental techniques to determine recombination efficiency and validate specific knockdown of *Pkd1* within lymphatics and/or endothelium. These experiments are ongoing in our laboratory. Instead, in our paper, we have selected to study the *Pkd1^RC^*mouse model, which mimics the temporal progression of human PKD and the kidney lymphatics of which not been studied until now.

Could the authors provide co-localization of lymphatic markers with VEGF-C? It would add considerably to the manuscript if kidney VEGF-C expression was also shown in development in relation to lymphatic development. This would be of particular interest in the *Pkd1*^*RC/R*C^ vs *Pkd1^+/+^*mice and should increase our understanding of why lymphatic vessel development is stunted in polycystic kidney disease. This might be possible on existing tissue collected from mice? Is VEGF-C required for the development of kidney lymphatics?

We completely agree that an evaluation of VEGF-C in both kidney development and PKD would be informative. Indeed, our previous work demonstrated that mice homozygous for a hypomorphic *Pkd1* allele have reduced *Vegfc* mRNA in the early stages of cystogenesis (Huang et al., 2016). However, our attempts at localising VEGF-C in kidney development using immunostaining or the *Vegfc^LacZ^* reporter mouse were thus far unsuccessful due to poor quality staining and a failure to obtain time-mated embryos at the required stages, respectively.

We have therefore examined whether VEGF-C promotes kidney lymphatic development. To do this, we isolated and cultured CD-1 mouse embryonic kidneys at E14.5 with media with or without 40ng/ml of recombinant VEGF-C. After two days in culture, we used 3D imaging and found aggregates of PROX1^+^/LYVE-1^+^ cells, the abundance of which were significantly increased when VEGF-C was added to the culture medium (please refer to Figure 2E and Figure 2—video 2 and accompanying figure legends and paragraph three of subsection “Characterization of conserved lymphatic endothelial cell clusters in the developing mammalian kidney” in the revised article). Therefore, although the cellular sources of VEGF-C during kidney development remain an open question, this ex vivo experiment provides strong evidence that VEGF-C is involved in renal lymphatic development and adds a functional element to the paper.

The authors should add a summary of the use of the different lymphatic markers and what they each may or may not relate to.

We have expanded the key resources table in the revised manuscript providing a summary of the lymphatic markers used, including the structures they label in lymphatic endothelial cells as well as information on other cell types which have been reported to express these markers in the developing kidney.

The lymphatic vessel diameter, branch points and volume between *Pkd1^+/+^*and CD-1 mice at the same time in development (E18.5) seem very different. Even more so than between *Pkd1^RC/RC^*and *Pkd1^+/+^* mice. The authors should explain what is the explanation for these differences. Some discussion about differences in lymphatic development between mouse strains should be added.

We thank the reviewers for raising this important point. We agree that these differences reflect differences in the temporal progression of lymphatic development in different mouse strains. We have added text in subsection “Defective renal lymphatic vascular architecture in polycystic kidney disease” to highlight this point. In prior reports knocking out key transcription factors or micro-RNAs governing lymphatic development, there are strain-dependent differences in phenotypes, such as subcutaneous edema, hemorrhage and embryonic lethality (François et al., 2008; Kontarakis et al., 2018). As all quantitative analyses described in our paper only compare between mice on the same genetic background, the strain differences do not impact on the interpretation of our comparisons.

[Editors' note: further revisions were requested prior to acceptance, as described below.]

The manuscript has been improved but there are some remaining issues that need to be addressed before acceptance.In the rebuttal, the authors note: "However, our attempts at localising VEGF-C in kidney development using immunostaining or the *Vegfc^LacZ^* reporter mouse were thus far unsuccessful due to poor quality staining and a failure to obtain time-mated embryos at the required stages, respectively." This analysis should be completed, as it is necessary to support the claim that VEGF-C is responsible for lymphatic vessel growth in the kidney, when it could be VEGF-D instead. VEGF-C needs to be expressed (nearby) to make the claim in the manuscript.

We appreciate the favourable evaluation of the Editors and the reviewers to our previous revision and the opportunity to address the remaining issues. In this amended version of the paper, we have performed the requested analysis of VEGF-C in kidney development to support the claim that VEGF-C is responsible for lymphatic vessel growth in the kidney.

To do this, we have examined the developing kidneys of *Vegfc^LacZ^* mice (Karkkainen et al., 2004). Wholemount X-gal staining of E15.5 kidneys from heterozygous embryos (*Vegfc^LacZ/+^*) demonstrated β-galactosidase (β-gal) activity in a branching pattern, mimicking the appearance of the renal arterial tree. Sections of X-gal-stained E16.5 *Vegfc^LacZ/+^* kidneys showed β-gal activity to be restricted to interstitial cells beneath the pelvis and adjacent arterioles. We further stained these sections for LYVE-1 and observed lumenized LYVE-1^+^ vessels in the hilum surrounded by β-gal-expressing interstitial cells. Together, these findings convey that the renal hilum; where kidney lymphatics first arise, is a VEGF-C-rich niche which likely promotes renal lymphangiogenesis.

Our new data is presented in Figure 1—figure supplement 4 and outlined in the main text.